# Controllable Fabrication and Oil–Water Separation Properties of Polyethylene Terephthaloyl-Ethylenediamine-IPN-poly(N-Isopropylacrylamide) Microcapsules

**DOI:** 10.3390/polym15010053

**Published:** 2022-12-23

**Authors:** Meng Liu, Dan Zhao, Hui Lv, Yunjing Liang, Yannan Yang, Zongguo Hong, Jingxue Liu, Kang Dai, Xincai Xiao

**Affiliations:** 1School of Pharmacy, South-Central University for Nationalities, Wuhan 430074, China; 2National Demonstration Center for Experimental Ethnopharmacology Education, South-Central University for Nationalities, Wuhan 430074, China; 3The College of Art and Science, The Ohio State University, Columbus, OH 43210, USA

**Keywords:** monodispersity, size repeatability, emulsion, microfluidic, poly(N-isopropylacrylamide)

## Abstract

In this paper, we report a microcapsule embedded PNIPAN in P (TPC-EDA) shell and it can be regarded as an interpenetrating polymer network (IPN) structure, which can accelerate the penetration of oily substances at a certain temperature, and the microcapsules are highly monodisperse and dimensionally reproducible. The proposed microcapsules were fabricated in a three-step process. The first step was the optimization of the conditions for preparing oil in water emulsions by microfluidic device. In the second step, monodisperse polyethylene terephthaloyl-ethylenediamine (P(TPC-EDA)) microcapsules were prepared by interfacial polymerization. In the third step, the final microcapsules with poly(N-isopropylacrylamide) (PNIPAM)-based interpenetrating polymer network (IPN) structure in P(TPC-EDA) shells were finished by free radical polymerization. We conducted careful data analysis on the size of the emulsion prepared by microfluidic technology and used a very intuitive functional relationship to show the production characteristics of microfluidics, which is rarely seen in other literatures. The results show that when the IPN-structured system swelled for 6 h, the adsorption capacity of kerosene was the largest, which was promising for water–oil separation or extraction and separation of hydrophobic drugs. Because we used microfluidic technology, the products obtained have good monodispersity and are expected to be produced in large quantities in industry.

## 1. Introduction

Emulsion occupies an important position in industries such as food [1,2,3], pharmaceutical [4,5,6,7], cosmetics [8,9,10], paint [11,12], and polymer materials [13,14]. In the above-mentioned various applications, the size and monodispersity control of the emulsion are particularly important. As a new drug-carrying system or other field developed in recent years, microcapsules have the functions of encapsulating slow-release and controlled-release drugs. It improves the stability of the drug in the body, prolongs the action time, reduces the number of administrations, and even improves the bioavailability [15]. The monodispersity and size of the microcapsules can precisely manipulate its distribution in the vessel and the drug release kinetics. Therefore, controllable fabrication for the microcapsules has both scientific and technological interest.

Generally speaking, the microcapsules prepared by the classical emulsification method have a relatively wide size distribution [16]. To control the monodispersity and size, some strategies are still continuing. Wang et al. [17] used microfluidic to prepare a novel core-shell APSI ultra-thin film hybrid capsule, which is highly monodisperse and has an average particle size of about 3.34 mm.

Microfluidic is a technique for manipulating tiny volumes of liquids [18,19]. The basic principle of droplet generation is to select two immiscible liquids: one is the continuous phase and the other is the dispersed phase [20]. The dispersed phase is separated into droplets of tiny volume by the shear stress of the continuous phase in the microchannel. The operating diagram is shown in Figure 1. After the dispersed phase enters the channel, the continuous phase pours in at the outlet of the channel, forming a shear force, thereby cutting the dispersed phase into a microsphere. In theory, it is possible to continue to extend backward to obtain multi-layered microspheres. The microfluidic device just fits the principle of the emulsion-template method, and the microfluidic technology can precisely control the size of the emulsion, thereby obtaining controllable size microcapsules. Since its development, microfluidic technology has been widely used in material synthesis [21,22] and single-cell analysis [23,24,25].

Due to its lower critical solution temperature (LCST) being 32–33 °C, the temperature-sensitive poly(N-isopropylacrylamide) (PNIPAM) presents a thermal response transition between hydrophilicity and hydrophobicity. When the ambient temperature is higher than its LCST, PNIPAM undergoes a phase transition from hydrophilic to hydrophobic, thus allowing oily substances to pass through [26], which was used to oil separation or extraction of the hydrophobic drugs. However, pure PNIPAM microcapsules are easy to collapse and not to restore its original globular shape [27], leaving little room for oil or other hydrophobic substance. By adding silica nanofillers to PNIPAM, Beata et al. [28] greatly improved its performance, and the pore walls of PNIPAM were more stable and less prone to collapse. The temperature response of the gel can change by 70% within 5.5 s, achieving a symmetric reversible volume response to temperature jumps. The pH-induced deswelling is still very fast (75 s for 70% of process). They thought the gels might be of interest as soft mechanical actuators.

The emulsion-templated method is a common one to prepare microcapsules. Patra et al. [29] used Pickering emulsion as a template to prepare stimuli-responsive microcapsules by molecular recognition of β-CD and adamantane (ADA) at the toluene–water interface. Using the formed O/W emulsion as a template, they undergo a polymerization reaction on the oil–water interface to form a polymer capsule. Therefore, the size of the microcapsules is controlled by the size of the O/W emulsion.

In this project, a self-made microfluidic device was used, and polyethylene terephthaloyl-ethylenediamine microcapsules [30] were prepared by the emulsion-templated method and interfacial polymerization [31], then a IPN-structured PNIPAM in the microcapsules surface was fabricated to avoid pure PNIPAM shells collapsing. As shown in Figure 1, the main content consists of the following parts. Firstly, the condition for preparing O/W emulsions by microfluidic device was optimized, the relationship between the parameters of the microfluidic device and the size of the emulsions was fit. Secondly, P(TPC-EDA) microcapsules were fabricated, and these microcapsules were characterized under SEM and XPS. Thirdly, the IPN-structured microcapsules were prepared. Finally, both of their swelling and oil absorption performance experiments were monitored. This study has certain implications for demulsification, oil pollution control, and absorption or loading of hydrophobic drugs.

## 2. Materials and Methods

### 2.1. Materials

N-isopropylacrylamide (NIPAM), Henan Dongchuang Chemical Products Co., Ltd. (Zhenzhou, China), was purified by n-hexane dissolution, filtrated, and recrystallizated at 4 °C. Arowana soybean oil was edible grade, Yihai Kerry Foodstuffs Marketing Co., Ltd. (Shanghai, China). Sodium dodecylbenzenesulfonate (SDS), Sinopharm Chemical Reagent Co., Ltd.(Shanghai, China), was chemically pure. Terephthaloyl chloride (TPC), Ron’s reagent (Shanghai, China), was analytical grade. Anhydrous ethylene glycol amine (EDA), Sinopharm Group Chemical Reagent Co., Ltd. (Shanghai, China), was analytically pure. Isopropanol, Sinopharm Group Chemical Reagent Co., Ltd. (Shanghai, China), was analytically pure. Ethyl acetate, Sinopharm Group Chemical Reagent Co., Ltd. (Shanghai, China), was analytically pure. Ammonium persulfate (APS), Sinopharm Chemical Reagent Co., Ltd. (Shanghai, China), was chemically pure. Kerosene, Ron Reagent (Shanghai, China), was reagent grade and used as received without any further purification. Well-deionized and deoxygenated water, whose resistance was larger than 16 MΩ, was used in all the synthesis processes.

Some instruments used included TSD01 syringe pump (Baoding Leifer Fluid Technology Co., Ltd., Baoding, China), freeze dryer (Beijing Boyikang Experiment Instrument Co., Ltd., Beijing, China), X-ray photoelectron spectrometer (XPS, Thermo Fisher Scientific Co., Ltd., Shanghai, China), field emission scanning electron microscope (SEM, SU8010, Hitachi Ltd., Tokyo, Japan), inverted biological microscope and its supporting software MvImage vt (Ningbo Sunny Instrument Co., Ltd., Ningbo, China), disposable intravenous infusion tube (Wuhan Wangguan Medical Equipment Co., Ltd., Wuhan, China), and SHA-C water bath constant temperature oscillator (Tianjin Sateris Experimental Analysis Instrument Manufacturing Factory, Tianjin, China). Both flat needle such as No. 21 (outer is 0.81 mm, inner is 0.51 mm), No. 30 (outer is 0.31 mm, inner is 0.13 mm) needle, and 5 mL and 20 mL BD syringe were used.

### 2.2. Preparation and Characterization of O/W Emulsion

#### 2.2.1. Preparation of O/W Emulsion

A 5 wt% SDS aqueous solution was configured, and a self-made microfluidic device was assembled. Soybean oil was used as the dispersed phase (referred to as the first phase), and the SDS aqueous solution was used as the continuous phase (referred to as the second phase). They were inserted into syringes, and then installed on syringe pumps. The flow rate of the second phase (*U*_2_) was fixed at 100 and 300 μL/min, the flow rate of the first phase (*U*_1_) was adjusted from 0 to 10 μL/min, and the emulsion was received after a period of stability. Then, the flow rate of the first phase was fixed at 5 μL/min, and the flow rate of the second phase was adjusted from 100 to 500 μL/min, and the same operation was performed.

#### 2.2.2. Measurement of O/W Emulsion Radius

After stabilizing for a period of time, the emulsion was taken. Then, their morphologies were observed under a microscope, and their radius were measured by the software MvImage vt matched with the microscope. Finally, the relationship between the two-phase flow rate and the radius was fitted.

#### 2.2.3. Calculation of the Monodispersity of the O/W Emulsion

Its monodispersity was characterized according to the following formula.
(1)$$\delta =\frac{D_{90}-D_{10}}{D_{50}}$$

The *δ* represents the monodispersity. *D*_90_, *D*_50_, and *D*_10_ represent the radius corresponding to 90%, 50%, and 10% of the cumulative distribution curve of the emulsion, respectively. The smaller the value of *δ* (*δ* < 0.4), the better the monodispersity.

#### 2.2.4. Repetitive Experiment

To verify the repeatability of the test, the same experiment was repeated three times according to the previous experimental conditions.

### 2.3. Preparation of the Microcapsules and the Final Samples

#### 2.3.1. Preparation of P(TPC-EDA) Microcapsules

According to the previous mature conditions for emulsion preparation, 5 wt% EDA aqueous solution, 5 wt% SDS aqueous solution, and TPC oil solution were prepared. TPC oil solution was used as the first phase, the flow rate was adjusted to 5 μL/min. The SDS aqueous solution was the second phase, and the flow rate was adjusted to 350 μL/min. The EDA aqueous solution was placed in the receiver. When the emulsion dropped into the EDA aqueous solution for several minutes, it could form white microspheres. Then, the microspheres were washed, soaked, and washed with isopropanol and ethyl acetate to remove the soybean oil inside the microspheres, and the obtained microspheres were freeze-dried to obtain dry microcapsules.

#### 2.3.2. Preparation of P(TPC-EDA)-IPN-PNIPAM Microcapsules

P(TPC-EDA) microcapsules were added to 0.5% wt NIPAM solution, 1% wt APS to NIPAM was added, nitrogen was introduced for 20 min, and then the microcapsules were placed in a 60 °C water bath. After 72 h, the microcapsules were taken out, washed with water, and then freeze-dried in vacuum for 24 h to obtain dry P(TPC-EDA)-IPN-PNIPAM microcapsules.

Table 1 showed the prescription of synthetic microcapsules, which were composed of TPC and EDA, and are listed in a table for convenience

### 2.4. Characterization of the Samples

#### 2.4.1. Microscopic and SEM Characterization of the Samples

The morphology and structure of all microcapsules obtained in the previous step were observed under an inverted optical microscope and SEM (SU8010, Hitachi Ltd., Tokyo, Japan).

#### 2.4.2. XPS Tests of the Microcapsules before and after Adding PNIPAM

In order to characterize whether PNIPAM was added, the dried microcapsules before and after adding PNIPAM were taken and tested by XPS (Thermo Fisher Scientific Co., Ltd., Shanghai, China) to analyze the elements.

#### 2.4.3. FTIR Tests of the Microcapsules before and after Adding PNIPAM

A small amount of P(TPC-EDA) microcapsules, PNIPAM gel, and P(TPC-EDA)-IPN-PNIPAM microcapsules were freeze-dried, then they were mixed and compressed with potassium bromide at a ratio of 1:100. Necolet Fourier transform infrared spectrometer was used for infrared spectroscopy analysis.

#### 2.4.4. The Swelling Recovery Experiment

A total of 2000 dry P(TPC-EDA) and P(TPC-EDA)-IPN-PNIPAM microcapsules were weighed. Then, these microcapsules were divided into 8 groups: 1 h, 2 h, 4 h, 8 h, 12 h, 24 h, 36 h, and 48 h. Next, 3 mL purified water was added in them and they were soaked at room temperature. The water outside the microcapsules were sucked away by absorbent paper after the corresponding time, and then they were weighed again. Finally, the degree of swelling was calculated according to the following formula.
(2)$$q=\frac{m_{1}}{m_{0}}$$
where the q represents the degree of swelling, *m*_1_ represents the weight of water added when swelling for a specified time, and *m*_0_ represents the weight of water when fully swollen

#### 2.4.5. Absorbing Kerosene Experiment

The dry P(TPC-EDA) and P(TPC-EDA)-IPN-PNIPAM microcapsules were divided into 2 h, 4 h, 6 h, 8 h, 10 h, 12 h, and 14 h. The microcapsules were soaked in room temperature water (less than the LCST) for the corresponding time, then they sucked away the surface liquid and were weighed, and then soaked in 40 °C kerosene (higher than the LCST) for 24 h after the microcapsules sucked off the kerosene on the surface and were weighed.

#### 2.4.6. Calculating the Internal Volume

The internal volume of the microcapsules was calculated from the increased weight of the dried microcapsules, according to the following formula.
(3)$$V=\frac{m_{1}}{\rho_{1}}+\frac{m_{2}}{\rho_{2}}$$
where *m*_1_ and *m*_2_ represent the increased weight of water and kerosene, respectively. *ρ*_1_ and *ρ*_2_ represent the density of water and kerosene, respectively.

## 3. Results

### 3.1. Characterization Results of O/W Emulsion

#### 3.1.1. Relationship between O/W Emulsion Radius and the Flow Rate

Figure 2 shows some microscopic images of the emulsion. As the flow rate of the first phase decreases, the emulsion size decreases significantly.

Figure 3 shows the repetitive results and the relationship between O/W droplet size and the flow rate of the different phase. From Figure 3a–c, every group has the same tendency, which shows that the fabrication had repeatability and controllability. As shown in Figure 3a, when the second phase flow rate was fixed at 100 μL/min, and the first phase flow rate was from 1 μL/min to 10 μL/min, the radius of emulsion increased with the increase of the flow rate of the first phase.

As shown in Figure 3b, when the flow rate of the second phase was fixed at 300 μL/min, and the flow rate of the first phase was from 1 μL/min to 10 μL/min, it was similar to the previous groups.

Figure 3c shows that the flow rate of the first phase was fixed at 5 μL/min, and the flow rate of the second phase was from 100 μL/min to 500 μL/min. With the increase of the flow rate of the second phase, the shear force on the first phase increases, so the radius of emulsion decreased with the increase of the flow rate of the second phase, and it also has a linear relationship.

As shown in Figure 3d, when the flow rate of the first phase was below 1 μL/min, the radius of the emulsion became unstable and irregular. We judged that the flow rate of the first phase was too low, but the flow rate of the second phase was too high, and the flow field at the intersection of the first phase and the second phase was unstable, resulting in uneven shear of the emulsion.

Figure 4 shows the relationship between the two-phase flow velocity (*U*_1_, *U*_2_) and the radius (R) by python. As shown in the figure, both the radius and the flow rate of emulsion change regularly within a certain range, and a plane was obtained by fitting the functional relationship between *U*_1_, *U*_2_, and *R*, which shows that the fabrication had repeatability and controllability again.

#### 3.1.2. Monodispersity of O/W Emulsion

Table 2 shows the δ of the emulsion at the flow rate from 1:300 μL/min to 10:300 μL/min and from 5:100 μL/min to 5:500 μL/min. It can be seen from the table that the results were basically around 0.100 (δ < 0.4), which indicates that the monodispersity of the emulsion was very good and stable, and the self-made microfluidic equipment used to prepare emulsion had great advantages over the traditional preparation methods.

### 3.2. Characterization of the Samples

#### 3.2.1. Micrographs and SEM Images of the Samples

Figure 5 shows the characterization of the microcapsules. Figure 5a shows the P(TPC-EDA) microcapsules under the optical microscope. It can be seen that the size was very uniform except for individual damage. Figure 5b shows the P(TPC-EDA) microcapsules under the SEM. The damaged microcapsules were caused from the brittleness of P(TPC-EDA) and compressed air during the sample preparation. However, their size uniformity can still be seen clearly. As for the appearance of the inner cavity of P(TPC-EDA) microcapsules, due to the fact that the volume of the internal oil phase was much smaller than that of the water phase during the interfacial polymerization of emulsion, the concentration of TPC in one of the reactants was very high, which made the polymer grow inward, so its interior became this irregular structure. Figure 5c shows that the surface of the latter was smoother than that of Figure 5d. We believed that PNIPAM gel filled the pits or holes on the surface of microcapsules. From the wall thickness before and after the addition of PNIPAM of the microcapsules in Figure 5e,f, it can be seen that the microcapsule wall thickness increased after the addition of PNIPAM.

#### 3.2.2. XPS Characterization of Samples

Figure 6 shows XPS results before (Figure 6a) and after (Figure 6b) adding PNIPAM. We judged according to the change of element content on the surface. As shown in Table 3, the content of N element increased after PNIPAM was added. We preliminarily judged that PNIPAM was added to the surface of P(TPC-EDA) microcapsules.

#### 3.2.3. FTIR Results of Samples

Figure 7 is the infrared spectrum of each sample, 1636 cm^−1^ is the stretching vibration absorption peak of amide I band C = O, 1545 cm^−1^ is the in-plane bending vibration peak of amide II band -NH-. Since the main functional groups of the three samples are all amide groups, there are characteristic absorption peaks of amide groups in all three samples. At 1375 cm^−1^ and 1385 cm^−1^ are the absorption peaks of isopropyl, these two peaks appear in the two samples of PNIPAM gel and P(TPC-EDA)-IPN-PNIPAM microcapsules, but not in P (in the samples of TPC-EDA) microcapsules; this represents the combination of PNIPAM and P(TPC-EDA) microcapsules.

#### 3.2.4. The Swelling Recovery Ratio of the Samples

Figure 8 shows that the recovery ratio of the samples with and without PNIPAM under soaking for 1 h, 2 h, 4 h, 8 h, 12 h, 24 h, 36 h, and 48 h, respectively, which was calculated according to Formula (2). The recovery results show that the swelling recovery degree of the dried microcapsules was close to 100% after 48 h and both changes were similar; the reason was that the mass ratio of PNIPAM was low. After about 15 h, the recovery speed with PNIPAM was more rapid than that of without PNIPAM.

#### 3.2.5. The Absorbing Kerosene Ability of the Samples

Figure 9a shows the absorbing weight of kerosene after the dry P (TPC-EDA)-IPN-PNIPAM and P (TPC-EDA) microcapsules swelling under the corresponding time. Compared with the absorbing kerosene of P (TPC-EDA), P (TPC-EDA)-IPN-PNIPAM clearly had high absorbing ability. When swelling for 6 h, the absorbing kerosene was the maximum because PNIPAM lost its own property after drying, including the conversion of hydrophobicity. After swelling for 6 h, the ability gradually recovered. At the same time, the microcapsules absorbed lower water, so it caused the absorbing kerosene to be up to the maximum. After 6 h, more and more water was absorbed, there was little space left for kerosene, so the absorbing kerosene can be reduced. As shown in Figure 9b, the liquid volume inside the microcapsules was calculated according to Formula (3). The changes of the liquid volume inside the microcapsules of the blank group and the experimental group are basically in the same trend, and remain basically constant after 6 h. The deviation in the results belongs to systematic error and is within our acceptance range. There will be breakage or loss, which is also inevitable. According to Formula (4), the volume of the microcapsule lumen should be 0.1256 mL. The internal liquid volume after swelling and oil absorption was a maximum of 0.1000 mL. The inner cavity utilization rate was about 80%. The above results indicate that the microcapsules can substantially maintain their shapes and provide a carrier for PNIPAM to exert its temperature-sensitive hydrophilic–lipophilic switching properties.

## 4. Conclusions

In this experiment, based on the principle of microfluidics and lotion template polymerization, the formation of lotion is precisely controlled by using a self-made microfluidics device, which can control the plane relationship between the droplet size and flow rate of continuous phase and discrete phase. After several groups of experiments, we obtained the quantitative relationship between the flow rate of continuous phase as well as dispersed phase of microfluidic and the droplet radius, and fit it into a binary function through python. The Var (R) of this binary function is 88.1054, which we believe meets the requirements of mass production in industry. Then, we synthesized composite microcapsules with good uniformity from TPC, EDA, and NIPAM. According to the temperature-sensitive characteristics of PNIPAM, the oil absorption experiment was carried out. It was found that after adding PNIPAM to P (TPC-EDA) microcapsule, and when the composite microcapsules swelled for 6 h, its ability to absorb kerosene is obviously improved. It has certain implications for the recovery of hydrophobic drugs and the treatment of environmental oil pollution.

## Figures and Tables

**Figure 1 polymers-15-00053-f001:**
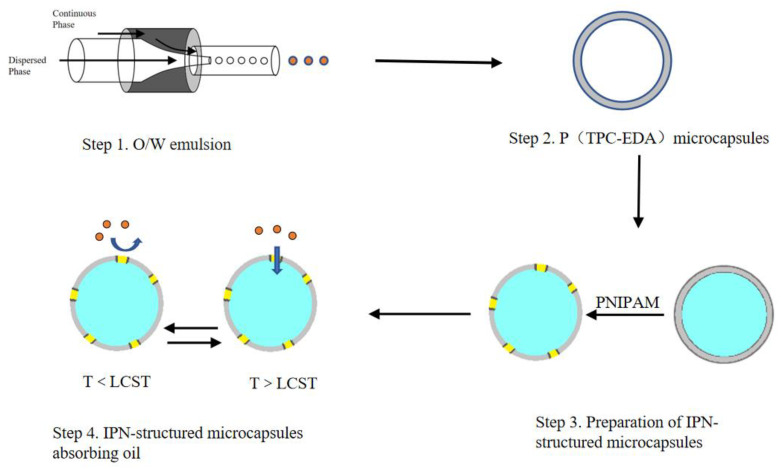
Experimental process of the microcapsules.

**Figure 2 polymers-15-00053-f002:**
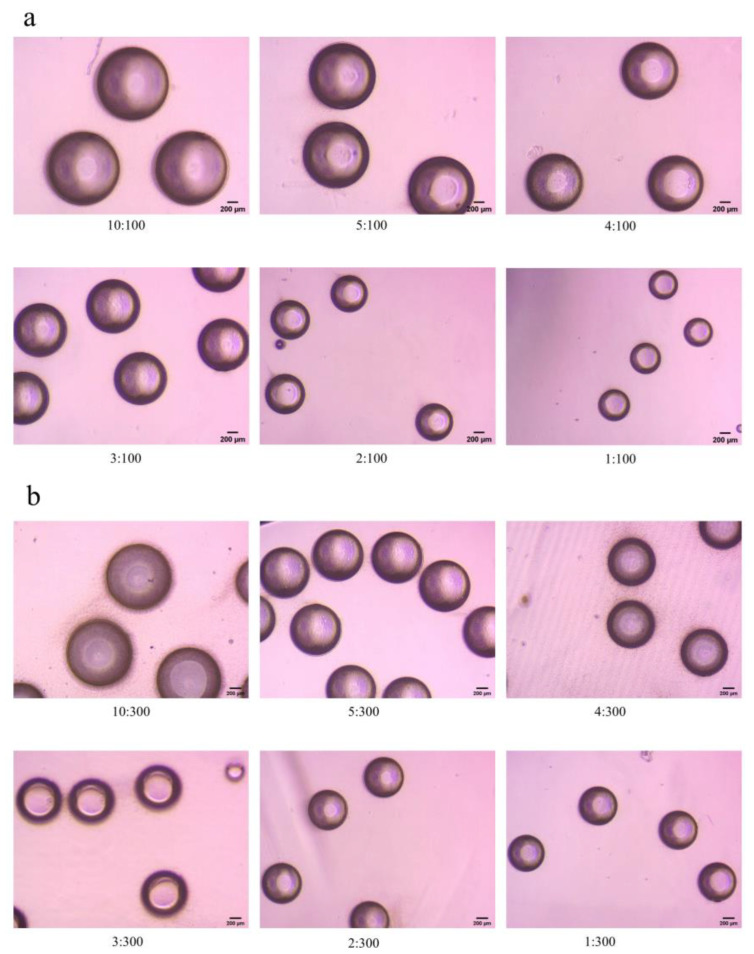
Microscopic images of partial emulsion (Scale bar 200 μm),(**a**) the second phase flow rate was fixed at 100 μL/min, and the first phase flow rate was from 1 μL/min to 10 μL/min, (**b**) the second phase flow rate was fixed at 300 μL/min, and the first phase flow rate was from 1 μL/min to 10 μL/min.

**Figure 3 polymers-15-00053-f003:**
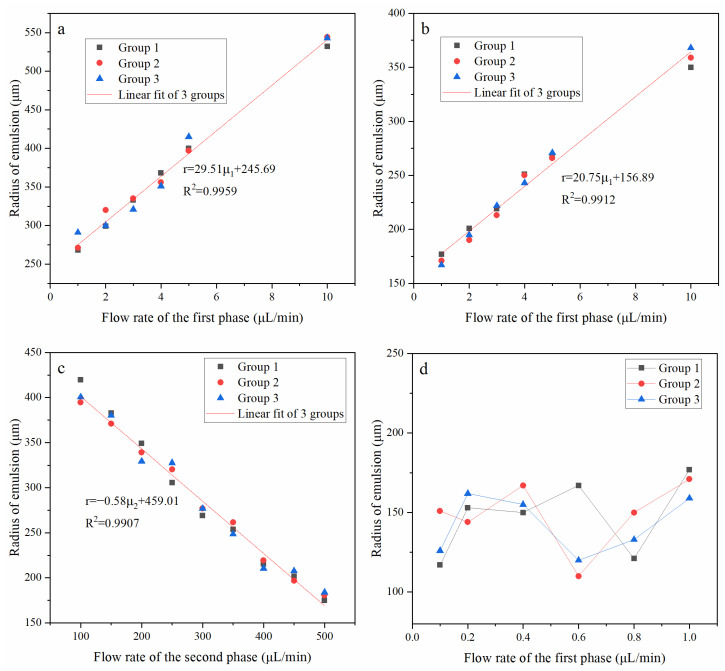
The repeatability and relationship between O/W droplet size and flow rate. (**a**) The flow rate of the second phase was 100 μL/min; (**b**,**d**) the flow rate of the second phase was 300 μL/min; (**c**) the flow rate of the first phase was 5 μL/min.

**Figure 4 polymers-15-00053-f004:**
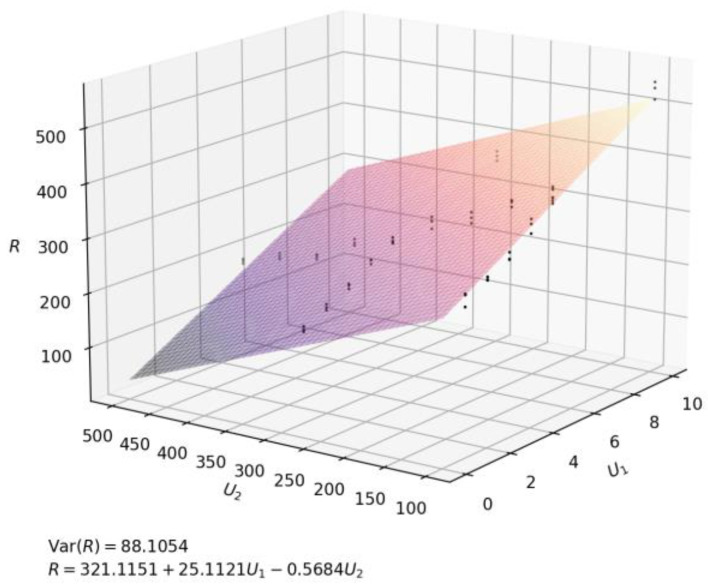
Fitting plane of *U*_1_, *U*_2_, and *R*.

**Figure 5 polymers-15-00053-f005:**
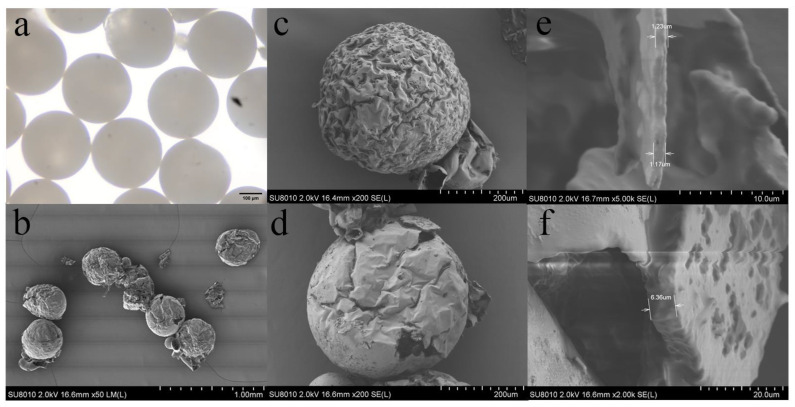
Morphology of P(TPC-EDA) microcapsules. (**a**) P(TPC-EDA) microcapsules under optical microscope (scale bar 200 μm); (**b**) P(TPC-EDA) microcapsules (scale bar 1 mm) and P(TPC-EDA) microcapsule lumen under SEM (scale bar 100 μm); (**c**) single P(TPC-EDA) microcapsule (scale bar 200 μm); (**d**) P(TPC-EDA) -IPN-PNIPAM microcapsules (scale bar 200 μm); (**e**,**f**) wall thickness of microcapsules before (scale bar 10 μm) and after adding PNIPAM (scale bar 20 μm).

**Figure 6 polymers-15-00053-f006:**
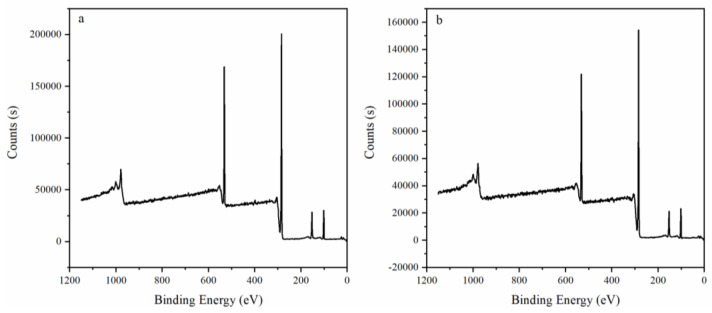
XPS results before and after adding PNIPAM. (**a**) Before adding PNIPAM. (**b**) After adding PNIPAM.

**Figure 7 polymers-15-00053-f007:**
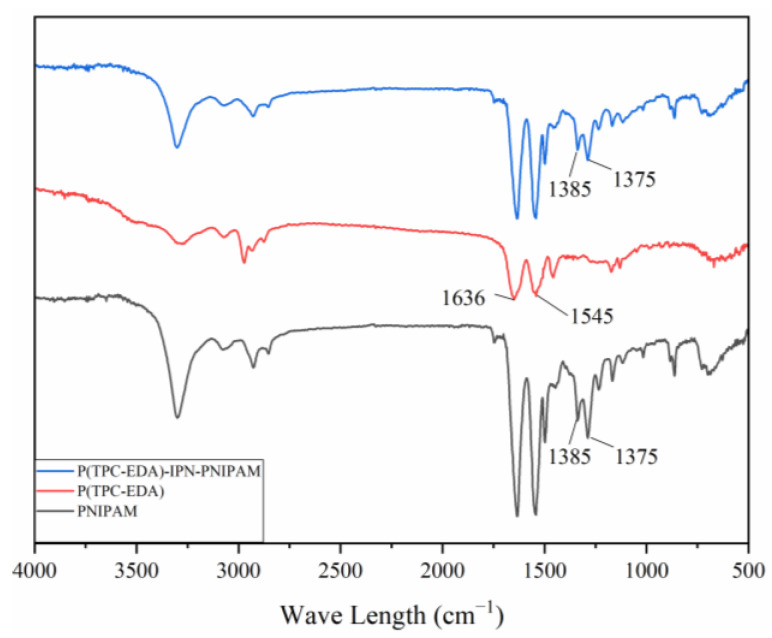
The FTIR spectrum of PNIPAM, P(TPC-EDA), and P(TPC-EDA)-IPN-PNIPAM.

**Figure 8 polymers-15-00053-f008:**
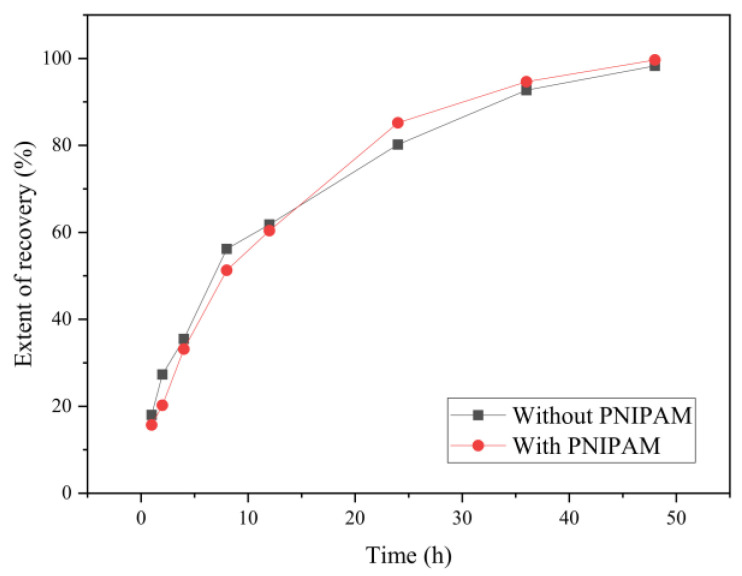
The swelling ratio of the sample adding before and after PNIPAM.

**Figure 9 polymers-15-00053-f009:**
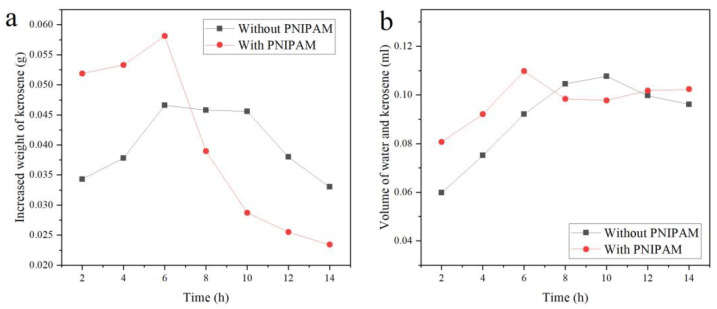
The ability of the samples absorbing kerosene. (**a**) Curve of oil absorption and swelling time. (**b**) Liquid volume curve of microcapsules after absorbing kerosene.

**Table 1 polymers-15-00053-t001:** Formulation of synthetic microcapsules.

	Concentration of Each Component	
Component Name	Without PNIPAM	With PNIPAM
TPC	5%wt	5%wt
EDA	5%wt	5%wt
NIPAM	0	0.5%wt

**Table 2 polymers-15-00053-t002:** Monodispersity and size of emulsion at each flow rate.

*U*_1_ (μL/min)	*U*_2_ (μL/min)	*R* (μm)	*δ*
1	300	172	0.116
2	300	195	0.114
3	300	218	0.116
4	300	248	0.106
5	300	268	0.116
10	300	389	0.091
5	100	405	0.126
5	150	378	0.124
5	200	339	0.081
5	250	318	0.100
5	300	274	0.096
5	350	255	0.114
5	400	215	0.112
5	450	202	0.133
5	500	180	0.093

**Table 3 polymers-15-00053-t003:** Changes of atomic content before and after adding PNIPAM.

Atomic Name	Content before Adding PNIPAM (%)	Content after Adding PNIPAM (%)
**C1s**	79.72	84.83
**O1s**	19.62	13.92
**N1s**	0.66	1.7

## Data Availability

Data presented in this study are available on request from the corresponding author.

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
