# Peer review of "Controllable Fabrication and Oil–Water Separation Properties of Polyethylene Terephthaloyl-Ethylenediamine-IPN-poly(N-Isopropylacrylamide) Microcapsules"

_polymers, 2022, doi:10.3390/polym15010053_

Round 1
Reviewer 1 Report
The manuscript entitled “Controllable fabrication and oil-water separation properties of polyethylene terephthaloyl-ethylenediamine-IPN-poly(N-iso- propylacrylamide) microcapsules” presents an IPN-structured microcapsule with thermosensitive properties, which can accelerate the penetration of oily substances at a certain temperature, and the microcapsules are highly monodisperse and dimensionally reproducible.
The current manuscript cannot be further considered for publication in polymers in its current status.
There are too many basic problems in the manuscript that have to be addressed:
1. Line 13 – IPN headline doesn’t have any explanation, the explanation is only in line 20.
2. Line 66 – there is name of paper of Beata at al. without the reference.
3. In Figure 1. the upper figure can’t be read.
4. In line 222 is says: “As shown in Fig. 3d, when the flow rate of the first phase was below 1μl/min…” while in Figure 3d the X axis says flow rate of the second phase.
5. The font of Figure 6 can’t be read at all.
6. In line 307 the word “The” is spare or should be placed differently.
7. Figure Legend of Figure 7. is lacking an explanation about the Figure.
Author Response
请参阅附件。

Reviewer 2 Report
The authors provided a method for emulsion formulation and characterized the formed emulsions. I suggest the author to add discussions on the scientific rationales behind the optimization.
Also, please make it more clear to the readers on the direct implications to scientific community, i.e. what is an optimal flow rate for certain applications?
It seems that a paragraph is missing on page 9 line 306.
Reviewer 3 Report
The manuscript titled " Controllable fabrication and oil-water separation properties of polyethylene terephthaloyl-ethylenediamine-IPN-poly(N-iso-propylacrylamide) microcapsules " describes the synthesis of IPN-structured microcapsule with thermosensitive properties. The manuscript is well arranged and in line with the polymers journal. The reviewer's comments are as follows:
1. Kindly, justify the use of kerosene.
2. Improve the figure captions.
3. Provide the composition of the microcapsules in tabular form.
4. Revise the conclusion. Conclude the manuscript by including the results obtained.
Round 2
Reviewer 1 Report
The manuscript entitled “Controllable fabrication and oil-water separation properties of polyethylene terephthaloyl-ethylenediamine-IPN-poly(N-iso- propylacrylamide) microcapsules” presents an IPN-structured microcapsule with thermosensitive properties, which can accelerate the penetration of oily substances at a certain temperature, and the microcapsules are highly monodisperse and dimensionally reproducible.
All comments have been properly addressed and therefore the manuscript can be further considered for publication in Polymers.
